# New Drug Development and Clinical Trial Design by Applying Genomic Information Management

**DOI:** 10.3390/pharmaceutics14081539

**Published:** 2022-07-24

**Authors:** Young Kyung Ko, Jeong-An Gim

**Affiliations:** 1Division of Pulmonary, Allergy and Critical Care Medicine, Department of Internal Medicine, Korea University Guro Hospital, Seoul 08308, Korea; youngsoka@naver.com; 2Medical Science Research Center, College of Medicine, Korea University Guro Hospital, Seoul 08308, Korea

**Keywords:** basket trial, clinical trial, genomic information, new drug development, umbrella trial

## Abstract

Depending on the patients’ genotype, the same drug may have different efficacies or side effects. With the cost of genomic analysis decreasing and reliability of analysis methods improving, vast amount of genomic information has been made available. Several studies in pharmacology have been based on genomic information to select the optimal drug, determine the dose, predict efficacy, and prevent side effects. This paper reviews the tissue specificity and genomic information of cancer. If the tissue specificity of cancer is low, cancer is induced in various organs based on a single gene mutation. Basket trials can be performed for carcinomas with low tissue specificity, confirming the efficacy of one drug for a single gene mutation in various carcinomas. Conversely, if the tissue specificity of cancer is high, cancer is induced in only one organ based on a single gene mutation. An umbrella trial can be performed for carcinomas with a high tissue specificity. Some drugs are effective for patients with a specific genotype. A companion diagnostic strategy that prescribes a specific drug for patients selected with a specific genotype is also reviewed. Genomic information is used in pharmacometrics to identify the relationship among pharmacokinetics, pharmacodynamics, and biomarkers of disease treatment effects. Utilizing genomic information, sophisticated clinical trials can be designed that will be better suited to the patients of specific genotypes. Genomic information also provides prospects for innovative drug development. Through proper genomic information management, factors relating to drug response and effects can be determined by selecting the appropriate data for analysis and by understanding the structure of the data. Selecting pre-processing and appropriate machine-learning libraries for use as machine-learning input features is also necessary. Professional curation of the output result is also required. Personalized medicine can be realized using a genome-based customized clinical trial design.

## 1. Background: Effective Clinical Trials Using Genomic Information

The human genome includes variants of DNA base sequences and epigenetic mutations, including changes in DNA methylation and histone acetylation patterns. Genomic variations, which include mutations in drug metabolism-related genes, can affect the pharmacokinetics, pharmacodynamics, efficacy, and safety of drugs [1,2]. This review describes the use of drug-related genomic information in drug development and clinical trial design. Personalized next-generation clinical trials, based on the individual genome, can be designed to maximize drug efficacy and minimize side effects.

With the rapid development of genome analysis technologies and computer performance, vast amounts of genomic information have been generated, which can be utilized in precision medicine. Representative cancer-related genomic information includes epidermal growth factor receptor (EGFR) mutation in non-small cell lung cancer (NSCLC) [3,4], ABL1 gene recombination in chronic myelogenous leukemia (CML) [5], and BRAF mutations in melanoma [6,7,8]. Genomic information related to cancer induction is available in databases, such as The Cancer Genome Atlas (TCGA) [9], Internal Cancer Genome Consortium (ICGC) [10], COSMIC [11], cBioPortal [12], OncoKB [13], MutaGene [14], and Cancer Genome Interpreter [15].

Open cancer genome data are generated using high-throughput technology. Cancer genome data published in TCGA include omics and clinical data of genomic variants, RNA-seq, and DNA methylation [9]. In cBioPortal, which is a rapid user-friendly platform, data on survival analysis related to variants, histological type, RNA-seq, or comparative analysis with methylation information are available [12]. The number of breast cancer patients is the largest in TCGA, in which variant information on 986 patients in 1097 clinical cases was reported. However, clinical data from TCGA do not include information about the drugs administered to the patients. There is no information on the effects and side effects of the prescribed drugs for each type of cancer. The effects of drugs can be estimated based on genomic information (mainly genomic variants) in individual clinical trials in “Drug Resistance” database [16] in COSMIC and CancerDR [17], and clinical trial results can be obtained from clinicaltrials.gov (accessed on 21 July 2022) [18].

The main objective of personalized medicine is to recommend treatment strategies and select drugs suitable for individuals based on their genomic information. Patient-specific clinical trials are necessary to realize the full potential of personalized medicine. Personal genomic information should be included in the eligibility criteria (EC) for clinical trials. The goal of this review is to provide indications on how to utilize genomic information in clinical trial designs and new drug developments. The points of this review can help prevent adverse drug reactions based on genetic information and find more effective patients.

## 2. Integrated Interpretation: Tissue Specificity and Environment of Cancer

Generally, carcinomas are caused by the accumulation of multiple genetic alteration in somatic cells, and tissue-specific frequencies of variants have been observed in various cancers. Tissue specificity of cancer is attributed to a variant of a specific cancer-related gene that causes organ-specific cancer [19,20]. Variants in cancer-related genes cause various types of carcinomas in different organs of the body. A genetic mutation in carcinomas with high tissue specificity results in the cancer occurring in a specific organ.

In carcinomas with low tissue specificity, a single gene mutation can cause carcinogenesis in various organs. The body consists of various cells, tissues, and organs, and each cell has the same genome sequence. The same genome sequence can perform diverse functions in different cells, depending on the changes in the epigenetic information and on various signaling mechanisms around the cell. Sensitivity to the cancer-causing factors also varies according to the cell type [19,21]. Thus, even in the presence of the same cancer-causing mutation, the probability of cancer occurrence may differ depending on the organ. Similarly, the tissue specificity of cancer can be explained by the representative examples listed below.

Mutations in the adenomatous polyposis coli (APC) gene are the cause of most familial adenomatous polyposis and colorectal cancer, but are rarely observed in other carcinomas [22]. Mutations in the Cadherin 1 (CDH1) gene are also a major cause of hereditary diffuse gastric cancer [23]. Mutations in the BRCA1 gene are mainly observed in carcinomas afflicting women, such as breast and ovarian cancer [24]. Typically, all patients with hairy cell leukemia (HCL) harbor variants in the BRAF gene [25]. Approximately 50% patients with melanoma and papillary thyroid carcinoma carry a variant of the BRAF gene [26,27]. On the other hand, approximately 10% of colorectal cancer patients harbor a variant of the BRAF gene [28].

In contrast, mutations in the TP53 gene confer low tissue specificity [29]. These mutations occur in most cancers, such as NSCLC, pancreatic ductal adenocarcinoma (PDAC), colorectal cancer, breast cancer, and ovarian cancer. The TP53 gene is involved in immune response and immunotherapy, and the wild-type p53 protein functions in mounting an adequate innate immune response. In cancer, mutant forms of the p53 protein act as a tumor antigen and induce a B-cell antibody response as well as a CD-8 killer T-cell response. In cancer immunotherapy, autoimmune and inflammatory responses, neurodegeneration, senescence, epigenetic instability, immune response, pathways, and therapeutic strategies targeting the TP53 gene and p53 protein have been discussed [29,30].

Tissue specificity of cancer is affected by various environmental factors such as metabolic abnormalities caused by diabetes or high blood pressure, infection (bacteria, viruses, parasites), and by immunocompetence [31,32,33]. These factors are classified as macroenvironments and microenvironments.

The tumor macroenvironment includes changes in body fat content, blood pressure, and blood sugar level caused by obesity, high blood pressure, and diabetes. A correlation between the tumor macroenvironment and the incidence of cancer has been reported. Pathologically and epidemiologically, the correlation patterns differ according to the type of carcinoma [34,35,36]. In diabetes, insulinemia and hyperglycemia are induced, initiating carcinogenesis. Insulinemia activates insulin-like growth factor signaling, and hyperglycemia supplies nutrients to the cancer cells, promoting acidification [33,36]. It also activates angiogenesis and cell proliferation signals via a chronic inflammatory response [36,37]. In obesity, an abnormal increase in the secretion of sex hormones derived from adipocytes, fibrosis of certain organs, and steatosis are also observed [38,39]. Furthermore, excessive cytokine secretion is induced owing to an abnormal inflammatory response, thereby increasing treatment resistance. Therefore, it is necessary to classify carcinomas according to the tumor macroenvironment influence.

The tumor microenvironment and oncogenic signaling are regulated by ligands that affect cell differentiation and receptors [40]. The organ-specific tissue differentiation is induced by stem cells in the adult tissues. This process is regulated by the epigenetic patterns of the cells constituting the tissues, self-renewal factors, and external factors [41]. Stem cell differentiation has different patterns depending on the tissue. Mesenchymal cells secret WNT proteins to maintain their stemness in the intestine. Epidermal interfollicular stem cells express their own WNT ligands for self-renewal [42]. Stem cells in colorectal cancers are maintained by secretion from activated myofibroblasts, whereas activated WNT-related signals accelerate cancer differentiation [43].

The cancer microenvironment can be explained by tumor heterogeneity. Cancer is a collection of malignant cells, cancer-associated fibroblasts, and tumor-associated macrophages, along with their ecosystem [44]. Cancer cells can be classified into infiltrating endothelial, hematopoietic, stromal, and other cell types, and their interactions have been studied [45,46]. Cancer cells evolved from a primary cancer, a concept known as cancer evolution [46,47]. Different cancer evolution patterns need to be observed accurately in individuals so that personalized treatment can be made available. Single-cell RNA-seq (scRNA-seq) is a technique that can be useful to better understand tumor heterogeneity. Specific gene expression levels for each cell type can be determined using scRNA-seq. Clustering and visualization techniques with dimensional reduction (t-SNE) for each cell type can be applied using scRNA-seq [48,49,50]. scRNA-seq has been applied in hepatocellular carcinoma [49], NSCLC [50], and primary breast cancer [48] and helps in the stratification and accurate classification of patients. This can maximize the sensitivity to appropriate drugs by understanding the pathways based on the status of the carcinoma.

The microenvironment of different cancers should also be considered in clinical trials. In hospitals, biopsies are performed on cancer patients, and are subjected to pathological analysis and genetic testing. Using a machine learning approach, patient information can be integrated to provide rapid and simple insights of clinical relevance. Owing to the increase in life expectancy and lack of exercise, complex variables related to chronic diseases affect the diagnosis and prognosis of cancer. Using a machine-learning approach, the patient information (genetic information, chronic disease status, and lifestyle) is pre-processed so that the machine-learning library can diagnose the disease. Then, cancer occurrence and prognosis-related factors can be processed using the machine-learning strategies, such as pattern recognition, classification, and visualization of results. Appropriate services have been suggested in clinical practice. Cancer type-wise genetic information is available in databases such as TCGA and COSMIC, and can be visually checked using cBioPortal [12], which is a user-friendly system, based on the patient information, providing an optimal treatment strategy is necessary. By analyzing these data, research and system developments can provide rapid and accurate insights for clinical decision-making.

## 3. Deposition, Application, and Indexing of Genomic Variation Information

In 2001, the Human Genome Project [51] resulted in the release of the human reference genome sequence, with new versions of the standard genome released in 2003, 2006, 2009, and 2013. Variants found in the standard genome and in other cancer patients have been stored in a database, with constant additions of information. Databases are used to predict cancer occurrence and to select a treatment strategy. For example, cancer genome projects such as TCGA [52] and ICGC [10] are publicly available, and the results of genome analysis for various diseases as well as cancer are deposited at National Center of Biotechnology Information (NCBI) Gene Expression Omnibus (GEO). However, these databases present challenges to clinical applications of datasets. To solve these challenges, web services that summarize the analysis tools and results related to cancer genomes have been developed. One of them, cBioPortal [12], allows users to search and analyze various cancer genome data in a user-friendly manner. Various data, such as genome, transcriptome, epigenome, and proteome data, obtained from cancer-derived tissues or cell lines, are collated and organized. These data are then annotated, curated, and indexed to allow researchers to analyze them. (Epi) Genetic changes according to cancer-derived samples, cancer-related genes, and signal transduction information can be observed, visualized, and linked to clinical information.

Distinguishing somatic and germline variants is important in identifying cancer-related variants. In germline variants, different patterns appear by race [53,54]. Therefore, for preventive medicine, it is critical to determine the proportion of cancer-related germline variants. Thus, in the 1000 Genomes Project, the ratio of variants by race was determined [55]. Associations between somatic and germline variants in several carcinomas have been confirmed using TCGA information [56,57,58]. This indicated the interaction of germline-somatic variants in tumorigenesis and assisted in understanding the mechanisms of cancer risk variants. The most representative cancer somatic variants database is COSMIC [59]. Most of the known mechanisms that induce carcinoma development include somatic variants. Since COSMIC was released in 2004, there have been rapid developments in next-generation sequencing (NGS) technology, computer analysis performance, and throughput. Although COSMIC has been modified for approximately 20 years, a reference database for appropriate comparisons with variants obtained in the clinical field is needed. Information exchange with external resources such as Ensembl, HGNC, and RefSeq is also insufficient; however, this is expected to be resolved by upgrading the annotation system [59]. The two-hit hypothesis and the optimal treatment strategy can overcome the limitations of COSMIC and maximize its advantages.

In the two-hit hypothesis proposed by Knudson in 1971, “in the dominantly inherited form, one mutation is inherited via germinal cells and the second occurs in somatic cells. In the non-hereditary form, both mutations occur in the somatic cells” [60]. There are continuing debates regarding the role of somatic and germline variants in the development of carcinoma. Whole-exome sequencing analysis of autism patients and their families revealed that the number of de novo variants in germline cells increased with age [61]. This was confirmed in a deCODE genetics study conducted in Iceland [62]. In summary, both somatic and germline variants play important roles in carcinogenesis.

An optimal treatment strategy is one that is based on the integrated information regarding the somatic and germline mutations, age, lifestyle, and the clinical information of the patient [63,64]. To develop a patient-specific somatic-germline variant-based treatment strategy, genomic information, along with patient information, must be collected longitudinally. The collection of the family history of the patient and testing for germline variants in the family are also important [65]. Recently, an integrated treatment strategy using machine learning and a prognosis prediction strategy were presented [66,67]. The efforts and databases to realize personalized medicine are described below.

The germline/somatic variant subcommittee, a multidisciplinary research committee of Clinical Genome Resource (ClinGen), was established in 2013 [68]. Somatic-germline zygosity is an algorithm for predicting the homozygous versus heterozygous variants and those of somatic versus germline origin, and was introduced by utilizing the sequence information obtained from the carcinoma samples. Modeling with allele frequency (AF), sequencing depth, tumor ploidy, and local copy number as inputs can assist in clinical decision making [69]. In another study, germline and somatic variants were analyzed and a model of cancer occurrence with age was generated, confirming that germline variants cause early-onset cancer, whereas somatic variants induce late-onset cancer [70]. For future generation of data that can be used for cancer diagnosis and clinical decisions, a two-hit strategy is needed to simultaneously analyze somatic and germline variants in tumor tissue and blood. Genomic data from various carcinomas or races have been collected that will serve as a basis for future cancer diagnosis and treatment strategies.

To select genomic variants for clinical trials, it is important to determine the ratio of variants by race. The cohort project carried out in several countries and the UK Biobank project are representative examples. Using the ratio of variants in each nation, it is possible to determine the variants crucial for the occurrence of a specific cancer. Large-scale cohort studies conducted in several countries have been listed in Table 1. Health and genetic indicators found in specific races for diseases, including cancer, can be explained using national variant data and risk factors. Rare variants showing a race-specific pattern can explain the genetic contribution to disease development, unlike common or de novo variants [71].

PharmGKB [72] and DrugBank [73] provide information relating to pharmacogenomics and the associations between known genotypes and drugs. These databases suggest that resistance and sensitivity are related to drug responses in clinical trials. Large volumes of genomic information related to drug responses have been produced and evidence for the clinical use of drugs has been presented [17,74,75].

Data on cancer-related gene expression and variants are provided in datasets such as the NCBI GEO and ArrayExpress. These data can be used preliminarily to identify variants and gene expression related to drug sensitivities and their side effects. For example, when “cancer drug sensitivity resistance” is queried in GEO, GSE102787 dataset is highlighted. Using GEO2R, researchers can select genes that are differentially expressed based on drug sensitivities. The omics information based on differences between two groups proposes the possibility of its clinical application.

To date, omics data based on the results of many clinical trials related to drug sensitivity and resistance have been published and further work is ongoing. Genomic and clinical data related to clinical trials are big data, and further processing is required to make clinical decisions under the regulation of bioethics laws. Moreover, appropriate indexing and cleaning processes for the stored and collected data are required. Thus, when a researcher uses the stored data, incorrect decisions can be prevented through excluding unnecessary or unstandardized data. In addition, data modeling and decision curation are required (Figure 1).

**Table 1 pharmaceutics-14-01539-t001:** Large-scale cohort cases for the realization of precision medicine by nation.

Nation	Project Name	Data Size	Link	Ref.
Multi-national consortium	1000 Genomes Project	4974 people	https://www.internationalgenome.org (accessed on 21 July 2022)	[55]
USA	Precision Medicine Initiative cohort program (All-of-Us)	1M people (To be completed in 2022)	https://allofus.nih.gov (accessed on 21 July 2022)	[76]
England	The 100,000 Genomes Project	5M people (To be completed in 2023)	https://www.england.nhs.uk/genomics/genomic-research/100000-genomes-project (accessed on 21 July 2022)	[77]
Iceland	deCODE Genetics	100K people (Completed)	https://www.decode.com (accessed on 21 July 2022)	[78]
Finland	FinnGen Research Project	50K (To be completed)	https://www.finngen.fi/en/for_researchers (accessed on 21 July 2022)	[79]
Korea	Korea Bio-resource Information System	500 people	https://www.kobic.re.kr/kobis (accessed on 21 July 2022)	[80]
Korea	Clinical & Omics Data Archive	780 people	https://coda.nih.go.kr (accessed on 21 July 2022)	[81]
Netherlands	The Genome of the Netherlands project (GoNL)	150K people (Completed)	https://www.nlgenome.nl (accessed on 21 July 2022)	[82]
Singapore	Genome Institute of Singapore (GIS)	1M people (To be completed in 2028)	https://www.a-star.edu.sg/gis (accessed on 21 July 2022)	[83]

## 4. Basket and Umbrella Trials

The development of genome analysis technology has enabled integrated analysis of various carcinomas. Advances in cancer research can now help to identify cancers with the shared biological mechanisms in different anatomical locations as well as cancers with different biological pathways in the same anatomical location. These advancements have transformed the paradigm that cancers originating from different anatomical organs have different biological mechanisms. Thus, the new cancer classification is based on molecular, cellular, and signaling mechanisms. Although derived from different anatomical organs, cancers with the same signaling mechanism may be subjected to the same treatment strategy. A drug modulating a specific signaling mechanisms can be applied to cancers originating from various anatomical organs that share that particular signaling mechanism; this clinical trial strategy is called a basket trial. Conversely, cancers originating from the same anatomical organ can be caused by different signaling mechanisms. Drugs that control different signaling mechanisms in the same cancer can be administered in a strategy, called an umbrella trial. These clinical trial strategies can help optimize the efficacy of new drugs (Figure 2).

In basket clinical trials, a single anticancer drug is tested against various carcinomas harboring the same genetic variant, whereas in umbrella clinical trials, several anticancer drugs are tested against a single carcinoma according to various genetic variants [84]. An example of a basket clinical trial is the phase II clinical trial of vemurafenib in 122 patients harboring a BRAF V600E mutation [85]. Vemurafenib, an inhibitor of BRAF V600 kinase, has been established to treat various carcinomas by an appropriate basket trial. Prior to the development of basket clinical trials, vemurafenib was effective in approximately 50% of patients with metastatic melanoma harboring the BRAF V600E mutation.

In various genomic information-based cancer studies, such as those based on TCGA, the BRAF V600E mutation was found in various cancers, such as NSCLC and colorectal cancer. Therefore, a basket clinical trial was conducted in patients harboring BRAF V600E mutations and cancers in tissues other than those of melanoma patients. This basket trial consisted of a total of 6 + 1 cohorts, with cohorts of patients with six types of cancer: NSCLC, ovarian cancer, colorectal cancer, cholangiocarcinoma, breast cancer, and multiple myeloma. The cancer progressed in the cohorts of patients with different types of cancer. Additionally, it progressed in patients with Erdheim–Chester disease and Langerhans cell histiocytosis. The results showed that the efficacy of vemurafenib was not the same in all cancers, with a response rate of 42% in NSCLC and 43% in Erdheim–Chester disease or Langerhans cell histiocytosis.

Unlike basket clinical trials, umbrella clinical trials test various treatment methods on the same carcinoma. Umbrella clinical trials can screen various treatment methods for a patient group or carcinoma for which there is no clear biomarker.

## 5. Companion Diagnosis: From the Genomics Point of View

Companion diagnosis (CDx) is a diagnostic method or diagnostic tool that is a “companion” for selecting disease-causing factors for targeted therapy [86]. Only diagnostic methods permitted by regulatory agencies can be used for specific targeted therapeutics. Clinical validity of the diagnostic and treatment methods used in CDx must be confirmed through clinical trials [87]. The CDx cases are presented in Table 2.

To use diagnostic tools for specific therapeutic agents, clinical evidence for interpreting diagnostic results must be considered [86]. Evidence-based recommendations are available for selecting drugs and clinical methods. The technical and environmental factors of different laboratories must also be correlated [88]. In 2014, the US Food and Drug Administration presented guidelines for mandatory CDx when developing targeted therapies. Similarly, in 2015, the Korean Ministry of Food and Drug Safety announced the ‘Guidelines for Approval and Review of In Vitro Companion Diagnostic Devices’.

Various factors can induce carcinoma formation, such as breast cancer, colorectal cancer, lung cancer, stomach cancer, pancreatic duct cancer, and melanoma. However, the same factors (EGFR or TP53 genes) also cause carcinoma in various organs [3,4,29,30]. Therefore, if the underlying mechanism is the same, a common therapeutic strategy can be used.

In the 1970s, the therapeutic effect of tamoxifen (Nolvadex), a breast cancer therapeutic agent, varied depending on the status of the estrogen receptor (ER) expression in patients with breast cancer. In the 1980s, it became known that the therapeutic effect on breast cancer varied depending on the HER2 gene mutation. Trastuzumab (Herceptin), a HER2 antagonist, was developed in the 1990s. As the therapeutic effect differs among patients depending on their genotype, considering patient genotype while selecting a specific drug and establishing a treatment strategy has attracted attention. In the 2000s, the research findings on signal transduction of cancer-causing factors were evaluated, and drugs that inhibit mutation-induced cancer-causing factors were developed. Representative drugs include gefitinib (Iressa) and erlotinib (Tarceva), which inhibit EGFR signaling, and imatinib (Gleevec), which is used for CML treatment [89]. These targeted anticancer agents selectively detect and inhibit specific targets expressed in the cancer cells. Therefore, the therapeutic effect is improved with reduced side effects (Figure 3).

In the 2010s, CDx was used for the development of an immune checkpoint inhibitor. Ipilimumab (Yervoy), approved as the first immune checkpoint inhibitor in 2011, inhibits the activity of CTLA-4, which is expressed on the surface of T cells and suppresses their function. In the mid-2010s, drugs inhibiting PD-1, which plays a similar role as CTLA-4, were developed. Pembrolizumab (Keytruda) and nivolumab (Opdivo) selectively inhibit PD-1 in NSCLC and melanoma. These immune checkpoint inhibitors maximize T cell activity by inhibiting suppressing molecules, such as CTLA-4, PD-1, and PD-L1. In the case of pembrolizumab and nivolumab, health insurance is offered in Korea if patients having stage IIIB or higher disease test positive for PD-L1 expression and who have not responded to previous platinum-based chemotherapy without treatment with a PD-1 inhibitor (Figure 3).

The targeted cancer drugs discussed in this paper act only on cancer cells with specific biomarkers, and if used in individuals without the specific targets, they can cause side effects. Therefore, a process of detecting a specific target based on the patient’s genetic or clinical information is necessary. In this case, the regulatory body must approve the process of screening the specific target.

To promote CDx, each entity involved in new drug development requires that pharmaceutical companies need to personalize clinical trial designs based on the patients’ genotypes. Diagnostic companies will need to discover factors related to cancer-causing signaling from the results of basic science research, such as cellular- and molecular-level signaling mechanisms, and design a method to rapidly and accurately screen them. Regulators will need to strengthen the supervision, direction, and guidance of efficient and safe patient-specific clinical trial designs. Health insurance entities will also need to optimize an appropriate fee for diagnosis and examination to use a specific drug and pay according to efficacy. CDx can present clinical evidence for the use of drugs for a specific target and can increase cancer treatment efficiency by applying personalized treatments to patients. Additionally, it can contribute to the financial security of the National Health Insurance by reducing the indiscriminate or incorrect use of targeted anticancer drugs.

**Table 2 pharmaceutics-14-01539-t002:** Cases of companion diagnosis.

Gene/Protein	Anticancer Agent	Indications	Biomarker	Routine Testing	Ref. Papers	Ref. CT
ALK	Crizotinib, ceritinib, alectinib, lolatinib, brigatinib	NSCLC	ALK translocation	FISH, IHC	[90,91]	NCT00932451
AR	Abiraterone, enzalutamide, dalurotamide, apalutamide	Prostate cancer	AR expression	IHC	[92]	NCT02485691
BCL-2	Venetoclax	CML	BCL-2 protein expression, BCL-2 amplification/translocation	IHC (FISH, DNA/RNA sequencing), PCR	[93]	NCT03552692
BCR/ABL	Imatinib, dasatinib, nilotinib, bosutinib, ponatinib	CML	BCR/ABL1 fusion	IHC, PCR, DNA sequencing	[5]	NCT00070499
BRAF	Dabrafenib+trametinib, vemurafenib+cobimetinib, encorafenib+binimetinib	Melanoma, NSCLC, ATC, HCL	BRAF V600E/K mutations	IHC, PCR, DNA sequencing	[6,7,8]	NCT01597908
C-KIT, PDGFR	Imatinib	GIST	c-KIT Exon 9 and 11 mutations, PDGFR mutations	IHC, DNA sequencing	[94]	NCT00117299
PDGFRB	Imatinib	Myelodysplastic/myeloproliferative syndromes	PDGFRB rearrangement	FISH	[95]	NCT00038675
BRCA	Olaparib, talazoparib, rucaparib	Breast cancer, ovarian cancer, prostate cancer	Germline/somatic BRCA 1/2 mutations	DNA sequencing	[96]	NCT03286842
CTLA-4	Ipilimumab	Melanoma		DNA sequencing, PCR	[97]	NCT01216696
ER/PR	Tamoxifen, raloxifene, fulvestrant, toremifine	Breast cancer	ER/PR expression	IHC	[98,99]	NCT00066690
erBB2/HER-2	Trastuzumab, pertuzumab, ado-trastuzumab, emtansine, neratinib	Breast cancer, gastric cancer	HER-2 protein expression, HER-2 amplification	IHC, FISH	[100]	NCT01702558
EGFR	Gefitinib, erlotinib, afatinib, dacomitinib	NSCLC	EGFR exon 19 deletion, exon 21 L858R mutation	DNA sequencing, PCR	[4]	NCT01955421
	Osimertinib		EGFR T790M mutation		[3]	NCT02474355
FGFR2/3	Erdafitinib	Bladder cancer	FGFR3 mutations, FGFR2/3 fusions	DNA sequencing, FISH	[101]	NCT05052372
FLT3	Midostaurin, gilteritinib	AML	FLT3 mutations	DNA sequencing, PCR	[102]	NCT04027309
IDH1/2	Ivosidenib, enasidenib	AML	IDH1/2 mutations	IHC, DNA sequencing	[103]	NCT02632708
MET	Crizotinib	NSCLC	MET amplification, MET exon 14 alterations	FISH, DNA/RNA sequencing	[104]	NCT00585195
MSI-H or dMMR	Pembrolizumab	MSI-H or dMMR solid tumors	MLH1, MSH2, MSH6, PMS2 protein expression, MSI high	IHC, DNA sequencing, PCR	[105]	NCT04082572
	Nivolumab and ipilimumab	Colorectal cancer			[106]	NCT04008030
NTRK	Larotrectinib, entrectinib	Solid tumors with NTRK fusions	NTRK protein expression, NTRK fusion	IHC, FISH, DNA/RNA sequencing	[107]	NCT02576431
PI3KCA	Alpelisib	Breast cancer	PI3KCA mutation	DNA sequencing	[108]	NCT02437318
PI3KCA (alpha and delta)	Copanlisib	FL	PI3K mutation	DNA sequencing	[109]	NCT01660451
PI3K (delta and gamma)	Duvelisib	CLL, SLL	PI3K mutation	DNA sequencing	[110]	NCT01476657
RAS	Cetuximab, panitumumab	CRC	KRAS/NRAS wildtype	DNA sequencing	[111]	NCT04117945
RET	LOXO-292	NSCLC, MTC	RET fusion, RET mutation	FISH, DNA/RNA sequencing	[112]	NCT03157128
ROS1	Crizotinib, entrectinib	NSCLC	ROS translocation	FISH, DNA/RNA sequencing	[113]	NCT04603807

FISH: Fluorescence in situ hybridization, ISC: Immunohistochemistry, NSCLC: non-small cell lung cancer, CML: Chronic myeloid leukemia, ATC: anaplastic thyroid cancer, HCL: hairy cell leukemia, GIST: Gastrointestinal stromal tumor, AML: Acute myeloid leukemia, MSI-H: Microsatellite instability high, dMMR: DNA mismatch repair deficiency, CRC: Colorectal cancer, MTC: medullary thyroid cancer, FL: Follicular lymphoma, CLL: Chronic lymphocytic leukemia, SLL: small lymphocytic lymphoma.

## 6. Genomic Information and Pharmacometrics

Pharmacometrics identifies and predicts the relationship among pharmacokinetics, pharmacodynamics, biomarkers, and therapeutic properties through mathematical and statistical models. The interaction between drugs and patients is quantitatively analyzed by constructing and simulating a mathematical model to assess the effects of treatment and adverse effects according to drug concentration. This is intended to accurately identify the drug exposure–response relationship of individuals and groups by reflecting individual differences, intra-individual variability, and various errors.

In econometric pharmacology, parameters relating to pharmacology, physiology, and pathology are used. Recently, genotypes or epigenotypes have also been included as parameters. Systems pharmacology analyzes the diversity of individual drug responses by synthesizing these parameters, thus enabling a holistic approach to determine drug responses by parsing the various elements constituting individual drug responses. The systems pharmacology was developed by the following three factors: the increasing number of samples with well-analyzed patient characteristics, the development of omics technology, and the increasing analysis networks based on omics data.

Genotyping of high-throughput sequencing results obtained using DNA chips or NGS techniques is necessary for subjects participating in clinical trials. If the phenotype is considered safe and efficacious, the related genotype should be extracted, thereby elaborating on the ramifications of the patient group according to genotype. For statistical processing and machine learning analysis, patient-specific labeling should be accurately performed, and the individual characteristics of the patients should be well described. Recent evidence suggests that both genetic and epigenetic factors, such as gene expression and DNA methylation, are related to drug responses. Correlation, eQTL, and multi-omics approaches can be used to extract relevant parameters related to drug response.

Attempts to incorporate genotypes into drug response modelling are ongoing. In a clinical trial of simvastatin, modeling using seven genotypes known to be related to drug metabolism was attempted [114]. Modeling was attempted using the change in DNA methylation level caused by the EGFR inhibitor gefitinib as a parameter, and it was confirmed that epigenotypes, such as DNA methylation patterns, can also be the subject of modeling [115].

## 7. Challenge: Genomic Information Management

Peter Drucker said, “If you can’t measure it, you can’t manage it [116].” Currently, technologies that can measure genomic information have been developed [117]. Hospitals are accumulating patient-derived NGS data for the diagnosis and selection of appropriate drugs or treatments [118]. A system for the quality control of the measured results and the supervision of the regulatory agency on the results was established. Although the results of treatment progress along with patient-derived laboratory data are accumulated along with the genotype, it is necessary to establish a system for decision-making and selecting appropriate treatment strategies for specific diseases in actual clinical practice. An optimal management strategy that includes appropriate storage, indexing, and ethical considerations for accumulating genomic information is required. The topic of the “genomic information management” strategy requires further evaluation (Figure 4) [119].

The primary goal for implementing the genomic information management strategy is to obtain the necessary insights to perform research on existing public omics data (Table 3). Omics data, such as in NCBI GEO and ArrayExpress, require data analysis, visualization, and extraction insights. The web-based databases presented in Table 3 can be used, and the genome information can be appropriately managed using machine learning.

Machine learning is used to discover rules from data, recognize patterns, and classify them based on the characteristics of the data content [66,67]. In order for the machine-learning analysis library to recognize the data well, the structuring of the data (categorical, continuous, and ranked) and pre-processing should be performed initially. To extract factors relating to optimal clinical trials and the safety and efficacy of personalized drugs, a clear definition of the data structure is necessary. It is necessary to properly classify the genotype into categorical and phenotypic information on pharmacokinetic parameters, efficacy, and safety into categorical and continuous types and accurately predict the structure of the data to be performed.

Traditionally, hospitals have obtained clinical laboratory data and disease diagnosis results from patients. Recently, NGS-based genetic information from patients and image information, such as from PET and CT, have been stored in the hospital’s computer network in a common data model. These data are appropriate for selecting personalized medicine and patient-specific treatment strategies. However, the structure and characteristics of each data must be accurately understood and used as input features for the machine learning library. It is also necessary to select appropriate machine learning library inputs that present the optimal treatment strategy as the output. Patient information in hospitals is personal, and regulatory agencies and IRB reviewers must be confident that the research and clinical trial design protect patient privacy.

## 8. Perspectives and Conclusions

A large amount of genetic information can be quickly retrieved, and patient-derived clinical data can be stored in hospitals. Machine learning techniques are becoming more sophisticated for discovering combinations, recognizing patterns, and classifying clinical data. Computer performance and data storage are improving. These data can assist with developing new drugs and designing optimal clinical trials. In this review, new drug development and clinical trial designs using genomic information are discussed. The three most important points are as follows: firstly, the appropriate clinical data for analysis must be selected, and the structure of the data must be understood; second, a machine learning input feature and a machine learning library should be selected as inputs; third, appropriate curation of the output result is required.

In the future, hospitals will continue to accumulate patient-derived genomic and clinical data, and advances in computer performance and sophisticated machine learning libraries will continue. Collaborative research with research institutes and companies that can analyze the data accumulated in hospitals is necessary. Appropriate access to anonymized patient information and legal regulations and measures to protect patients’ personal information are required. Thus, patient-specific treatment will become increasingly sophisticated, the effects of treatment will increase, and the side effects of treatment will continue to decrease.

## Figures and Tables

**Figure 1 pharmaceutics-14-01539-f001:**
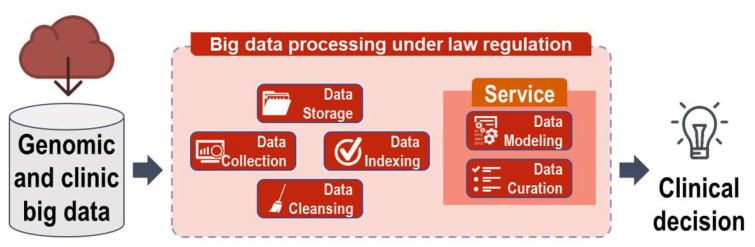
Processing strategy for genomic and clinical data. Data collected have to be stored, indexed, and cleaned for use at a later stage. Data modeling and curation are shown for the clinical decision system. All processes must be performed under the regulation of the bioethics law.

**Figure 2 pharmaceutics-14-01539-f002:**
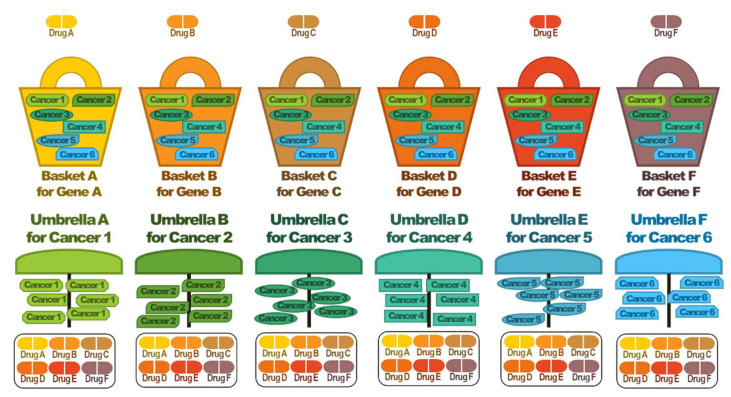
Basket and umbrella trials.

**Figure 3 pharmaceutics-14-01539-f003:**
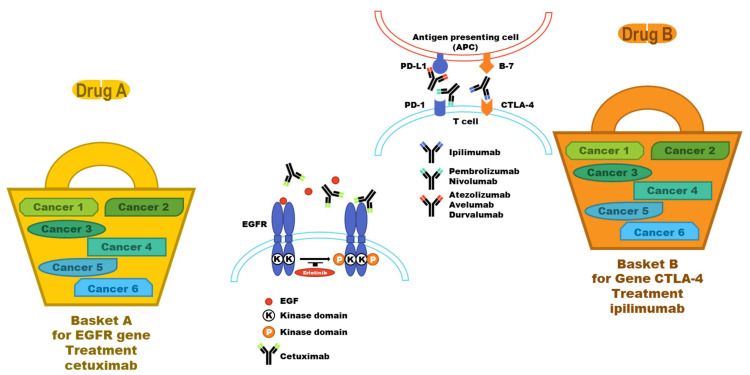
Examples of basket trials in EGFR.

**Figure 4 pharmaceutics-14-01539-f004:**
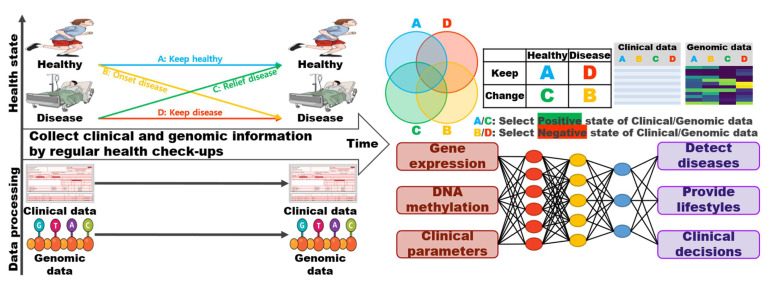
A comprehensive model of genomic information management with clinical data. Two omics data, gene expression and DNA methylation patterns could be changed by aging. The genomic data and clinical data of an individual are continuously collected over time. We aim to develop a model that can predict disease prediction, provide appropriate lifestyle habits, or present evidence that can be used in clinical practice by discovering genomic data that predicts changes in health status based on the collected data and applying machine learning to each data. Strategies for presenting insights based on patient-derived genomic information. Hospitals track and accumulate clinical information for chronic disease patients. Clinical information explains the maintenance of health, deterioration of the health state, and recovery of health over time. Integrate clinical and genomic information to find factors related to maintaining healthy states. The optimal combination is presented through machine learning, disease detection, lifestyle suggestion, and clinical decision basis.

**Table 3 pharmaceutics-14-01539-t003:** Cases of extracting new insights through public omics data.

Case	Query	Source	Output	Accessibility	Ref.
DBATE	Gene symbols	13 large RNA-seq from human healthy and disease tissues from NCBI GEO	Expression values that can be visualized in several ways	http://bioinformatica.uniroma2.it/DBATE (accessed on 21 July 2022)	[120]
MENT	Gene symbols or conditions of genomic data	NCBI GEO and TCGA	Patterns and gene list of DNA methylation, gene expression and their correlation in diverse cancers	https://mgrc.kribb.re.kr:8080/MENT/ (Unconnected)	[121]
GEM-TREND	Gene symbols	GEO, ArrayExpress, researchers’ websites	GEO series and platform ID, series title, similarity score, and *p*-value are displayed, network visualization	https://openebench.bsc.es/tool/gem-trend/ (accessed on 21 July 2022)	[122]
GeneXX	Gene symbols	NCBI GEO, transcriptome data	Stratified by exercise type, training status, sex, and time point postexercise	https://genexx.shinyapps.io/genexx (accessed on 21 July 2022)	[123]
GeneATLAS	GWAS catalog no.	UK Biobank	A large database of associations between hundreds of traits and millions of variants using the UK Biobank cohort	http://geneatlas.roslin.ed.ac.uk (accessed on 21 July 2022)	[124]
GliomaDB	Gene symbols	NCBI GEO, TCGA, CGGA, MSK-IMPACT, US FDA, PharmGKB of Genomic, transcriptomic, epigenomic, clinical information	Kaplan-Meier plot. The interactive heatmap visualization of the multi-omics data	http://cgga.org.cn:9091/gliomasdb (accessed on 21 July 2022)	[125]
Metamex	Gene symbols	Oligo package, limma package, DESeq2 package, NCBI GEO.	Skeletal muscle transcriptional responses to different modes of exercise and an online interface to readily interrogate the database	https://metamex.serve.scilifelab.se (accessed on 21 July 2022)	[126]
Oncopression	Gene symbols	NCBI GEO, ArrayExpress, ICGC, ExpressionAtlas, cBioPortal, ExAc Browser, oncomine (Rhodes)	Sample statistics of oncopression, Validity of dataset integration, Use of oncopression in cancer research	http://www.oncopression.com (accessed on 21 July 2022)	[127]
RefEx	Gene symbols, disease names	ESTs, Affymetrix GeneChip, CAGE, RNA-Seq, NCBI gene ID	Integration of publicly available gene expression data, visualize with BodyParts3D, extraction of genes with tissue-specific expression patterns, gene expression visualization of the FANTOM5 CAGE data	https://refex.dbcls.jp (accessed on 21 July 2022)	[128]
ReGEO	Gene symbols	GEO, NCBI, Search by keyword, GSE Accession, Pubmed ID, Experiment Type, Organism, Disease, Timepoints	Identify and categorize data for their integrative data analysis	https://regeo.org (accessed on 21 July 2022)	[129]

GEO: Gene Expression Omnibus. GEO series and platform ID are start as “GSE” and “GPL”, respectively.

## Data Availability

A link to the database presented in this manuscript is provided.

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
