# Peer review of "New Drug Development and Clinical Trial Design by Applying Genomic Information Management"

_pharmaceutics, 2022, doi:10.3390/pharmaceutics14081539_

Round 1
Reviewer 1 Report
The organization of the review is hard to follow. For example, in section 1, the last paragraph does not seem to follow logically from the previous ones.
Some sections (like sections 6 and 7) seem to be disconnected from the remaining ones. I believe the aim of the authors is to highlight several phases of drug development that can or should be influenced by patient genomic information if we wish to develop personalized medicine. But this is not very clear in the text.
In the beginning of section 2 the text seems to imply that cancers can be caused by single mutations, which is highly unlikely. Generally, cancer is caused by the accumulation of multiple genetic alterations in somatic cells.
Sentences like: "Single-cell RNA-seq (scRNA-seq) is a technique to explain tumor heterogeneity."(line 134) do not make sense. Eventually the authors mean something like: Single-cell RNA-seq is a technique that can be useful to better understand tumor heterogeneity". More sentences like this are spread throughout the manuscript. Careful language editing is needed because these sentences make the interpretation of the text very difficult.
Author Response
The organization of the review is hard to follow. For example, in section 1, the last paragraph does not seem to follow logically from the previous ones.
Response: Thanks for the point. Section 1 contains, and we tried to provide an overview that genomic information should be useful in clinical trials and new drug development. The first sentence of the last paragraph was deleted because it was logically awkward [Line 62].
Some sections (like sections 6 and 7) seem to be disconnected from the remaining ones. I believe the aim of the authors is to highlight several phases of drug development that can or should be influenced by patient genomic information if we wish to develop personalized medicine. But this is not very clear in the text.
Response: Section 6 is a brief introduction to modeling and pharmacometrics in drug development, and section 7 is the concept of genomic information management. Although it is a bit far from the overall content, it is briefly introduced because it is essential for new drug development and personalized medicine. If there is content that needs to be added, or if there is a point to be organized with a picture or table, please suggest it.
In the beginning of section 2 the text seems to imply that cancers can be caused by single mutations, which is highly unlikely. Generally, cancer is caused by the accumulation of multiple genetic alterations in somatic cells.
Response: We agree with the reviewer's opinion and added one related sentence to the beginning of the first paragraph of section 2 [Lines 72-73].
“Generally, carcinomas are caused by the accumulation of multiple genetic alteration in somatic cells, and tissue-specific frequencies of variants have been observed in various cancers.”
Sentences like: "Single-cell RNA-seq (scRNA-seq) is a technique to explain tumor heterogeneity."(line 134) do not make sense. Eventually the authors mean something like: Single-cell RNA-seq is a technique that can be useful to better understand tumor heterogeneity". More sentences like this are spread throughout the manuscript. Careful language editing is needed because these sentences make the interpretation of the text very difficult.
Response: We are not native English speakers, and received the English proofreading service of "Editage" (KOSCH_7620). Present "Certificate" as an attachment. After the correction, the sentence presented differently from our intentions was modified as a whole, but there may be parts of the text that are difficult. If you provide additional information on this, we will correct it. The presented sentence has been modified [Lines 137-138].
Reviewer 2 Report
Line 23: Utilizing genomic information, sophisticated clinical trials can be designed that will be better suited to the patients of specific genotypes.
Certainly, any effects or adverse events related to the genetic information would be useful information at the time of clinical trials. However, a trial that only proves an association with certain genetic information would require a large number of cases and would be inefficient. What does the author envision (or aim for) in writing this text? Please clarify whether you want to include genetic information in the final indication of the drug, to avoid adverse events by knowing the genetic information in advance, or to find effective cases.
Line 63-
I agree with your opinions.
Line 161
We now understand the possibility that the genetic and environmental information of cancer cells will become clearer, making it easier to predict prognosis. On the other hand, I believe that information from clinical trials will be essential to determining a treatment plan based on the genetic information of cancer. What kind of efforts should be made to address this issue in the future?
Line 266
I thought the explanation of the BASKET and UMBRELLA tests was easy to understand.
Line 405
Modeling was attempted using the change in DNA methylation level caused by the EGFR inhibitor gefitinib as a parameter, and it was confirmed that epigenotypes,
I think the effort to model the changes in DNA methylation levels caused by the EGFR inhibitor gefitinib as a parameter is very groundbreaking. On the other hand, how useful is this kind of modeling of epigenetic effects in other anticancer drugs? Is it possible to extrapolate the modeling to in vitro experiments? Can you discuss its versatility?
We believe that the challenges are very well summarized.
Author Response
Line 23: Utilizing genomic information, sophisticated clinical trials can be designed that will be better suited to the patients of specific genotypes.
Certainly, any effects or adverse events related to the genetic information would be useful information at the time of clinical trials. However, a trial that only proves an association with certain genetic information would require a large number of cases and would be inefficient. What does the author envision (or aim for) in writing this text? Please clarify whether you want to include genetic information in the final indication of the drug, to avoid adverse events by knowing the genetic information in advance, or to find effective cases.
Response: Thank you for your comment. The questions you have suggested are very critical and make the direction of this review more clearly. We emphasized clinical trial design for realizing personalized medicine in this review, which answers the second and third questions of the reviewer. In this regard, a description has been added as follows [Lines 67-69].
“The points of this review can help prevent adverse drug reactions based on genetic in-formation and find more effective patients.”
Line 63-I agree with your opinions.
Response: Thank you for your opinion. The sentence of line 63 is what we want to say in this review. The first sentence of the fourth paragraph in section 1 was deleted because it was logically awkward.
Line 161
We now understand the possibility that the genetic and environmental information of cancer cells will become clearer, making it easier to predict prognosis. On the other hand, I believe that information from clinical trials will be essential to determining a treatment plan based on the genetic information of cancer. What kind of efforts should be made to address this issue in the future?
Response: This is a very good point, and a schematic diagram is presented in Figure 1. Appropriate indexing and cleaning of the database in Table 1 are required, which are presented in section 3.
Line 266: I thought the explanation of the BASKET and UMBRELLA tests was easy to understand.
Response: Thank you for your opinion. Figure 2 is also a major point of this review.
Line 405: Modeling was attempted using the change in DNA methylation level caused by the EGFR inhibitor gefitinib as a parameter, and it was confirmed that epigenotypes,
I think the effort to model the changes in DNA methylation levels caused by the EGFR inhibitor gefitinib as a parameter is very groundbreaking. On the other hand, how useful is this kind of modeling of epigenetic effects in other anticancer drugs? Is it possible to extrapolate the modeling to in vitro experiments? Can you discuss its versatility?
Response: Corresponding author (J-AG) studied epigenotypes in the laboratory where I worked before, but the uncertainty was higher compared to the genotypes. So, statistical significance could not be provided. Therefore, it is considered as a long-term task and has not been dealt with in depth in this review.
We believe that the challenges are very well summarized.
Response: Thank you for your opinion. The authors are researchers working in hospitals, and have summarized our concerns.
Round 2
Reviewer 1 Report
I am happy with the authors reply.
Reviewer 2 Report
There is no comment.
The authors answered all queries.